# Understanding the Experience of Service Users in an Integrated Care Programme for Obesity and Mental Health: A Qualitative Investigation of Total Wellbeing Luton

**DOI:** 10.3390/ijerph19020817

**Published:** 2022-01-12

**Authors:** Fani Liapi, Angel Marie Chater, Julia Vera Pescheny, Gurch Randhawa, Yannis Pappas

**Affiliations:** 1Faculty of Health and Social Science, Institute for Health Research, University of Bedfordshire, Luton LU2 8LE, UK; gurch.randhawa@beds.ac.uk (G.R.); yannis.pappas@beds.ac.uk (Y.P.); 2Faculty of Education and Sport, Institute for Sport and Physical Activity Research, University of Bedfordshire, Bedford MK41 9EA, UK; angel.chater@beds.ac.uk; 3Patient Services, Biogen, Riedenburgerstraße 7, 81677 Munich, Germany; julia.pescheny@biogen.com

**Keywords:** evaluation, experiences, integrated care, mental health, obesity, service

## Abstract

Obesity is a complex public health issue with multiple contributing factors. The emphasis on joined care has led to the development and implementation of a number of integrated care interventions targeting obesity and mental health. The purpose of this study was to examine user experience in an integrated care programme for obesity and mental health in Luton, UK. Semi-structured interviews were conducted with a purposeful sample of service users (N = 14). Interview transcripts were analysed using thematic analysis. Analysis of the interviews identified six main themes for understanding service users’ experiences of integrated care: (1) ‘A user-centered system’, (2) ‘Supports behaviour change’, (3) ‘Valued social support’, (4) ‘Communication is key’, (5) ‘Flexible referral process’, and (6) ‘Positive impact on life’. These themes describe how the service is operated, evidence perceived value service users place on social support in behavior change intervention, and address which service areas work well and which require improvement. The findings of these interviews have offered a significant contribution to understanding what service users value the most in an integrated healthcare setting. Service users value ongoing support and being listened to by healthcare professionals, as well as the camaraderie and knowledge acquisition to support their own behaviour change and promote self-regulation following their participation in the programme.

## 1. Introduction

Obesity is a global public health priority [1]. The World Health Organization (WHO) states that worldwide obesity tripled between 1975 and 2016 [2]. In 2019 the Health Survey for England showed that 64.3% of adults were above a normal weight, with 35.6% of adults being overweight and 28.7% of adults obese [3]. Obesity and excess weight are associated with a large number of health problems, such as cardiovascular disease, type 2 diabetes, and cancer [4,5,6,7]. In addition, there is strong evidence in the literature that obesity is associated with several mental health conditions, such as depression and anxiety disorders [8,9,10,11,12,13].

In England, in 2019/20 alone, there were more than a million hospital admissions linked to obesity [14]. Obesity-related illness is a major cost to the UK National Health Service (NHS) in the United Kingdom. By 2050, it is projected that the NHS will be spending GBP 9.7 billion per year on treating obesity-related ill health [14]. Therefore, preventing and treating obesity represents a national priority to improve health and quality of life and reduce financial pressure on the NHS.

In response to the increasing prevalence of obesity in the UK, Public Health England and the NHS are focusing on ways of managing obesity effectively, and recognise the need for more integrated health care [15,16]. Local authorities and NHS clinical commissioning groups commission integrated care services that include behaviour change support to tackle obesity [15]. Integrated care systems are thought to be able to meet the increasing complexity of people’s needs in tackling obesity [14].

Total Wellbeing is an integrated care pathway in Luton, UK. Total Wellbeing was commissioned by the Luton Borough Council and NHS Luton Clinical Commissioning Group (NHS Luton CCG) in April 2018 to bring together social, physical and emotional wellbeing in one service. The service aims to support people to support themselves, with the use of evidence-based approaches to achieve sustainable behaviour change. It offers a wide range of programmes, such as mental health support, quitting smoking, weight management, social prescription, exercise on referral, and free NHS health checks.

A published review states that service users’ perceptions are rarely included in evaluations of integrated care services [17]. Understanding these experiences provides evidence on to the extent to which integrated care addresses the needs of the service users [18]. Consequently, service users’ narratives are vital in the evaluation of the service, and provide a valuable insight into how an integrated care programme for obesity and mental health works, in what areas the service demonstrates high quality, which areas may need improvement, and what users value the most about the care they receive.

As integrated care initiatives continue to develop, there is a need to better understand their impact on service users. The present study focused on weight management and Improving Access to Psychological Therapies (IAPT) programmes, as obesity and mental health are on the list of public health priorities in the UK [16]. The study aimed to examine service users’ experiences with Total Wellbeing Luton and their engagement with weight management and IAPT programmes, as well as to develop an understanding of how services are implemented. The findings may be valuable to service providers in improving existing services and maximizing expected impact on service users.

## 2. Methods

### 2.1. Study Design

A qualitative design used semi-structured interviews with service users to explore their experiences and perceptions of the integrated weight management service that they received from Total Wellbeing Luton. The present study was planned to begin in March 2020. On 11 March 2020, the WHO declared the COVID-19 outbreak as a global pandemic [19]. In the absence of any pharmaceutical intervention, the only strategy against COVID-19 was to reduce social contacts and engage in mitigation behaviours. On 23 March 2020, the UK government released social-distancing guidance to reduce the risk of transmission. The resulting social restrictions impacted the interview process by eliminating face-to-face interviews. Therefore, online interviews were found to be valuable for researchers conducting interviews during this time.

### 2.2. Study Sample

Purposive sampling was used in this study; participants were selected based on certain predefined criteria. To reduce recall bias, service users had to have used the services of Total Wellbeing Luton in the 12 months prior to the recruitment period (March 2020 to July 2021). In addition, participants were required to be able to communicate effectively in English.

The sample size in previous studies that aimed to explore the experiences and satisfaction of service users with health services ranged from 9–29 participants [20,21]. Based on the above, the planned sample size was estimated at between 13–20 service users, despite saturation being commonly used as a criterion when judging whether data collection should be continued or not in qualitative research [22]. Therefore, the recruitment of the participants stopped when saturation was reached.

Fourteen service users were interviewed, nine women and five men. Some had been referred to other services, including Improving Access to Psychological Therapy (IAPT) and social prescription (SP) (Table 1).

### 2.3. Data Collection

Individual semi-structured interviews were conducted (Appendix A: Interview schedule for service users). Due to the COVID-19 outbreak, the interviews were conducted through online communication platforms (Zoom, Skype, Microsoft Teams) or over the phone for social distancing purposes. Before the interviews, the participants were fully informed about the interview process. Participants were informed that the sessions would be audio recorded using the researcher’s personal audio recorder. The researcher highlighted that there was no use of cloud storage and the participants were reassured that there was no risk of being hacked. They were reassured that any information they shared would be kept anonymous. Written, informed consent for participation in the study was signed and emailed to the researcher by the participants prior to the interviews, and consent was confirmed verbally again before the interviews. Interviews were conducted following a semi-structured interview schedule. The length of the interview audio recordings were between seventeen and forty-five minutes. The interview schedule was developed after reviewing previous interview studies in health settings and our experience of working in integrated care and public health. The drafted schedule was discussed with the co-authors and funding partners, who provided valuable feedback.

### 2.4. Data Analysis

The present study aims to explore the lived experiences of the participants and interpret them. This is in line with the aim of interpretive phenomenology, that is, to uncover what a lived experience means to the individual [23]. Therefore, the research approach was informed by the principles of interpretive phenomenology. Phenomenology is a form of qualitative research that seeks to describe a phenomenon by studying the individual’s lived experiences [24]. Data saturation was reached in the examined sample.

The interviews were audio-recorded, transcribed verbatim, and analysed using thematic content analysis [25]. After interviewing each service user, a reflective note summary was documented based on the interview in order to contextualise the data. The reflexive approach promoted self-awareness regarding assumptions and preconceptions, and facilitated in-depth understanding. This process enhanced the validity of the research findings in the interpretation stage. The principal researcher (F.L.) transcribed the interviews in order to be immersed in the data and to develop a deep understanding of the context, as suggested by Braun and Clarke (2006). To ensure the principles of anonymity, any information that might reflect the identity of the participants were removed. Each participant was described as a ‘service user’ and assigned a number, e.g., service user 1. The transcripts were imported into the computer-assisted qualitative data analysis programme NVivo 11 (QSR International, Burlington, MA, USA to assist in the analysis of the data. F.L. conducted line-by-line coding by looking for patterns. The themes were derived from the data inductively according to Braun and Clarke [25]. Thirty percent of the transcripts were double-coded independently by J.V.P., and any discrepancies were discussed between the two coders. When the two coders agreed, the final definitions of the codes were assigned. The codes were revised by F.L, J.V.P., and A.M.C., then refined using a reflexive approach. Then, the codes were turned into themes. F.L. and A.M.C. discussed the identified themes and subthemes, and a coding matrix was constructed. This was mapped to with indicative quotes by F.L. and checked against the coding matrix by A.M.C. A further two iterations were made to ensure that the final thematic map represented the data, and consensus on themes was reached. A coding tree was produced to assist with visualisation of the findings (see Figure 1).

## 3. Results

The thematic analysis led to the identification of six main themes which describe the experiences of service users regarding the service they received from Total Wellbeing and indicate areas of success and areas for improvement. These main themes are: (1) a user-centered system, which (2) supports behaviour change, and provides (3) valued social support, (4) communication is key, and the service offers a (5) flexible referral process, and (6) positive impact on life. These core themes were structured into sub-themes to further describe the service users’ experiences and perceptions. A brief description of the core themes, subthemes, and supporting quotes are included in Appendix A.

### 3.1. Theme 1: A User-Centered System

Participants described a user-centred approach in which the service responded to their needs. According to service users, a user-centered system is one which provides a wide range of health support, both physical and emotional if required, and involves service users in decisions about their care plan.

#### 3.1.1. Offers Holistic Support

The data provided evidence that service users’ needs were identified by the healthcare professionals and support was offered if needed. Service users stated that they had been offered access to other health programmes through the service in addition to the one that they have been initially referred to after being holistically assessed by a health care professional.

*“They are there for everybody, and a wide range of help and services are available. Particularly now when you know things, over the last year or so with COVID-19, uncertainties and mental health problems, either to make you feel as though you’re not alone”.* (Service user 1)

*“The GP referred me to weight management programme, and then they referred me to social prescription”.* (Service user 6)

#### 3.1.2. Patient Activation

Patient involvement in care plans was highlighted with a sense of empowerment. The majority of the participants, nine out of fourteen, expressed that there was always the opportunity to be involved in their care plan, especially when they participated in the exercise sessions available.

*“They gave us an opportunity to discuss and share ideas, back and forth with each other. And that was quite good really. They didn’t dominate the class and gave us an opportunity”.* (Service user 5)

### 3.2. Theme 2: Supports Behaviour Change

Service users stated that they noticed changes to their health status as a result of their engagement in the weight management and mental health support programme. They highlighted knowledge acquisition, which helped them to make changes to their behaviour, maintain a healthy life, and self-manage their health. In addition, service users stressed that they were motivated to adopt healthier behaviours as a result of their participation in the programmes.

#### 3.2.1. Knowledge Acquisition

Service users emphasised that they were able to understand food labels and to identify what types of food were beneficial for them after the completion of the 12-week WM programme. They also learned techniques that helped them to manage their anxiety levels.

*“I was really interested in reading food labels, I had health problems, I had to go into FOD map diet. They really help with that. I learned how to read the food labels”* (Service user 14)

*“And the handouts that I was given, I still use today”.* (Service user 2)

#### 3.2.2. Increasing Motivation towards New Behaviours

Service users expressed that they faced difficulties in staying motivated and working out at home during the national lockdown, although on the other hand they expressed that when it was allowed they were more motivated to attend group exercise sessions. They also highlighted that peer support was an important factor which increased their motivation to attend exercise sessions.

*“If you’re doing it on your own, you have to have the motivation to do it. If you’ve got a class to go to, then, it’s in your schedule, and you will go. […] The physical benefit, but it’s also the social benefit as well, because there’s a good group of people there. […] I think because doing it in a group makes you go there. […] Monday morning, I go to a gym it’s a regular habit”.* (Service user 13)

One interviewee commented that the increased motivation to engage in physical activity led to working out in other places as well:

*“I am more motivated. I wish, I could go to the gym because while I’m not doing that…. […] Last night was nine o’clock, I played some music and I was dancing like crazy and sweaty, because I go to work out, and there’s nowhere to go… I did some exercise. You know, I think it made me feel more… to want to move on”.* (Service user 5)

#### 3.2.3. Enabling Self-Management

Service users who engaged with the IAPT service reported that they were able to understand patterns of thoughts or behaviours and work on them using self-help techniques. For example, one participant mentioned that keeping a journal was a helpful technique at the end of a stressful day. Participants highlighted that they do not feel “cured” but that they feel more confident to help themselves when they feel distressed due to the IAPT programme.

*“The therapist showed me different techniques, things like a journal. It’s called a worries diary, I read it every day, and if I ever have a stressful day I still go back to that. And so I still go back to all those still now”.* (Service user 3)

*“I am extremely happy with that service. […] I don’t feel I’m cured of it, but I certainly do feel more able to say, so much more, and know where to look for guidance if things are not heading in the right direction”.* (Service user 5)

### 3.3. Theme 3; Valued Social Support

Participants valued the ongoing support from healthcare professionals and the camaraderie they felt among fellow service users. Participants also raised the need for someone to follow up with them and check whether they had experienced adversity.

#### 3.3.1. Ongoing Health Professional Support

Service users valued the ongoing support they received from healthcare professionals during the intervention. Staff were available to be contacted by the service users even during atypical service hours. Participants appreciated that they were able to contact healthcare professionals easily and at any time to ask for advice.

*“It’s been lockdown and I’ve kind of gone off track. And, you know, he was always able to see and not just like a couple of minutes and say… “Do this, do this, do this”, it wasn’t like that, he gave me a lot of time, even on the phone”.* (Service user 6)

#### 3.3.2. Camaraderie among Other Service Users

Participants expressed that they valued the friendships that they built with other service users. They stated that the face-to-face sessions were an opportunity to get them out of the house, meet new people, and reduce feelings of isolation.

*“It was nice because I’m at home a lot. It is the first time that I’ve got to do something like this. Got to go in and work with like a group of people and get to meet new people, got to see some new faces and staff”.* (Service user 6)

*“The camaraderie. When I first came, we barely had coffee once every few weeks. And then we start coming down here to get a coffee, which is lovely. They lift you emotionally”.* (Service user 12)

Participants stressed the importance of the “family atmosphere” that they experienced from the Total Wellbeing staff and from staff at the delivery venue. They also highlighted that finding common experiences with their peers provided them a space to share their journey.

*“I think not only with the service you get from the staff, it is the people you’ve never known before. I have illnesses similar to yours. They’re making you to discuss, when you talk to people. That’s the most important part of everything. […] You could do the workouts at home, which we did with Joe Wicks, but wasn’t quite the same. […] So again, I go back to coming back to people, not only the staff, but the people you meet. It’s all about that. It’s a mixture of human connection. […] And I think they’ve got a great group of people here to try it the right way. […] I value the support from the people. They are here to advise, to look after”.* (Service user14)

#### 3.3.3. More Follow-Up Is Needed

Participants who completed the 12 sessions offered by the weight management and IAPT programme reported that they would like someone to follow up with them and keep them informed of their progress. They described the 12-week exercise plan as a “taster”, and mentioned that it would be beneficial for someone to follow up with them after six months to check on their commitment to behavioural changes.

*“And then there should be a report of your progress. Is it working or is it not working? To have measurements, is it working or is it not working? […] After the program, after six months down the line, they should ask “how are you getting on?” It is a tester. What happened after the tester? And you’ll always find some people. If you have 20, you might end up with five, who just want to keep going. Yes, it makes a difference to these five and should be worth it.”* (Service user 5)

*“At the end of the therapy, maybe someone to get like check-up on the people that are having therapy to make sure they’re okay or if like they need anything else. It would be a good idea.”* (Service user 2)

### 3.4. Theme 4: Communication Is Key

The service users’ narratives show that health care professionals were active listeners and responsive to the users’ needs. While the communication between service users and health care professionals was good, participants did express that some areas need attention.

#### 3.4.1. Active Listening and Responding to Needs

In all cases, the participants agreed that the healthcare professionals “really” listen to them. Participants felt comfortable expressing themselves to staff and also felt physically supported.

*“They listened to everything I said and they made me feel very comfortable to open up because it’s not something that I’ve been able to do before.”* (Service user 2)

*“I’ve told him that I wobble a lot, I don’t have good balance and he does help me and he does put things around to help me to do.”* (Service user 12)

Some expressed that they received support tailored to their individual needs, although they also appreciated the collective nature of the programme. They understood that a group programme has to meet the needs of different people. A feeling of “freedom” in a group programme was expressed which allowed service users to adapt the programme to their personal needs and goals.

*“I think I receive support based on my individual needs, but on others as well. […] I believe, they all know our needs.”* (Service user 14)

#### 3.4.2. Online vs. Face-to-Face

Because of the national lockdown due to the COVID-19 outbreak, Total Wellbeing made plans to run some services remotely. The weight management programme was one such programme. A service user who completed the online course stated that he intended to attend the face-to-face sessions as soon as it was allowed to run again. The participant explained that the face-to-face sessions allowed the development of conversations, which permitted direct answers to any questions he might have:

*“I think the online courses, it’s all right, and they give you information base. The online course doesn’t answer some of your questions. I mean you can talk to someone face-to-face and say, well, I’ve got this. And that can give you a direct answer rather than you’re thinking, well, is that right, is that wrong.”* (Service user 8)

#### 3.4.3. Unaware of Integration

Service users reported that they were not aware that Total Wellbeing is an integrated service which offers a wide range of support for physical and emotional health. They expressed their disappointment that nobody informed them about the available services, which would have been appreciated and useful.

*“I didn’t know it was a holist program. I think it would be nice that they explained what they were in the beginning. Saying, we could do this. Nobody ever said that to me. We do this, this, this, and this, you gradually find out for yourself really, just asking questions. I wasn’t aware of what they did until you graduate. It would be nice if you were sat down and someone spoke to you and told you: “We can help you completely.” No one ever said that; that would be nice and useful.”* (Service user 11)

#### 3.4.4. Poor Visibility of the Service

Participants expressed that the service had not been advertised enough and it was difficult for them to get enough information about the service and the available programmes. Some of the participants were proactive and were able to find the information that they needed; however, this was not the case for everyone. Service users suggested that the advertising strategy of the service should be improved:

*“I would say I’d highly recommend it. But it’s a lot of hard work in my heart to get the information. And because I am a person like that. I know how to get information. I am quite proactive in doing things. I could access it, but for a lot of people that would have been very difficult to do. […] I think when you get into it, it works. That’s just how you know it’s there…. for me the information of what they can do for you is not obvious.”* (Service user 11)

While service users were satisfied with the service and even recommended it to their friends, it is interesting that participants felt that some GP surgeries were not aware of Total Wellbeing or not informed enough to understand what programmes were available in the service or to refer patients. Participants reiterated that the service needs to be advertised widely:

*“I am speaking to a friend of mine, a couple of friends of mine and they were like saying you’re very lucky because they will also have health issues and their GP is very reluctant to refer them to this type of service. They’re not even aware of this kind of service, they don’t even know about it. They need to advertise a bit more.”* (Service user 6)

### 3.5. Theme 5: Flexible Referral Process

This theme concerns functional aspects of the service. It covers incoming referrals made by healthcare professional, self-referrals, and the accessibility of the service. Service users discussed how they were referred to the service, which is linked with the uptake of the programme.

#### 3.5.1. Health Professional Encouragement

Health care professionals seemed to be the initiators of participation in the weight management programme. Formal support from health professionals appeared to encourage uptake.

*“Because my doctor had concerns over my BMI. So, and he kept on saying to me, like you’ve got to do something, and then earlier this year I went to see. And he said to me, you’ve got to accept some help. So I said, All right, fair enough. So that’s what I came and I applied for the course, I got recommended for the course.”* (Service user 8)

#### 3.5.2. Ability to Self-Refer

Self-initiated uptake was reported by service users as well. After being informed about the available programmes and the option of self-referral to the service, service users decided to refer themselves to programmes that would help them to improve their health.

*“And then, after all that lockdown. I contacted the gym again. I’m a member there and I said, “Look, I really want to get back into exercise again and I need some help.” And so they gave me [name of member of staff], and she was brilliant. We met twice in the gym and we went to a program and then lockdown again. So, it has been wonderful. Really, really good.”* (Service user 11)

#### 3.5.3. Ease of Access

Overall, participants reported that access to Total Wellbeing was an easy process for them. Several contacted the service themselves, while others received a call from the service after a referral was made by a health care professional.

*“They had leaflets up on the wall. I just took one of those and phoned that number. No problems at all.”* (Service user 11)

*“They’ve got some nurses at the surgery that I use, and they put me forward to the nurse who then rang me and said, they’ve got someone to do with people losing weight and stuff like that. And they then recommended me and someone contacted me about it.”* (Service user 8)

### 3.6. Theme 6: Positive Impact on Life

The majority of the participants, thirteen out of fourteen, reported that they noticed positive changes in their physical and/or emotional health and social life after engaging with the programmes offered by Total Wellbeing.

#### 3.6.1. Improved Health and Wellbeing

Service users who engaged with the weight management programme noticed an improvement in their physical health, as they had lost weight and regulated their blood pressure.

*“I have. I managed to lose two stone, my blood pressure in normal now, the nurse said she was really shocked, that is beautiful, whatever you’re doing, keep it up. So, she was really good, I saw a lot of changes within myself.”* (Service user 6)

*“It made you think about what you want to eat, or what is helping you. I’m conscious of what I did, but it made me more aware of certain foods that I thought was ok. And in doing this, I lost weight over a week.”* (Service user 9)

#### 3.6.2. Changes in Social Life

Participants reported that they noticed changes in their social life after their engagement with Total Wellbeing programmes. They mentioned that they were happy to socialize, and did not avoid social opportunities as they used to:

*“I definitely go out and socialize more, and how I was feeling before, I think before I went out and it was make not to want to go. But now, I like what I’ve learned in my mind and make sure that I don’t avoid situations just because of how I’m feeling because I’m never going to overcome it if I keep avoiding them. Definitely, like, I’m more willing to go and socialize.”* (Service user 2)

#### 3.6.3. Optimistic Outlook on Life

Many participants expressed that their view on life had changed after their participation in Total Wellbeing programmes. They felt physically and emotionally better overall, and they reported that their lives were improved on the whole:

*“They helped me physically, mentally and emotionally. So I will say they changed my life. I was suffering from a lot of health problems and rapidly a lot of it improved.”* (Service user 6)

*“I’m literally like a holy person. I said what my main goal is to get back myself again. What happened is to find myself again.”* (Service user 3)

## 4. Discussion

This study explored the experiences and perceptions of service users participating in weight management and IAPT programmes delivered as part of an integrated care service called Total Wellbeing Luton. Our interview data generated six core themes: (1) “A user-centered system”, (2) “Supports behaviour change”, (3) “Valued social support”, (4) “Communication is key”, (5) “Flexible referral process”, and (6) “Positive impact on life”. The findings indicate several achievements of the service as well as potential areas for improvement.

Total Wellbeing Luton offers holistic and individualised support and promotes service users’ active involvement in their care plan. Participants stated that they had the opportunity to be involved in their care plan and that the sessions were not dominated by the healthcare professionals. The service users highlighted knowledge acquisition regarding self-management techniques and how to support their dietary needs after the completion of the programme. This is in accord with a systematic review which revealed evidence of service users’ positive views on the focus of weight management interventions providing dietary advice and promoting physical activity [26].

Past studies have reported that group-based interventions have been described by service users as an opportunity to socialize and have fun with their peers [27,28,29]. Data from the current study supports this notion. Although there is contradictory evidence in the literature which reports that service users experience embarrassment [27,30] and difficulties in discussing sensitive issues [27,31] in group-based sessions, service users of Total Wellbeing Luton did not report such negative experiences while being part of a group. In contrast, they reflected that group-based sessions encouraged opportunities for peer interaction with people with similar experiences. A systematic review provides further evidence that group-based interventions are valued by service users in that they create a space to share similar experiences [26]. This is a positive aspect of Total Wellbeing Luton to be celebrated.

Past literature documents that group interactions during a patient education programme can be more important for improving skills than the actual content of the programme [32]. The present study found that social interaction with people with similar experiences leads to the creation of new relationships and friendships, which are vital for emotional wellbeing. The feeling of togetherness increased users’ motivation to attend the group sessions, and thus their chances of continuing with healthy lifestyle changes. The findings of this study have several similarities with that of Solberg et al. (2014), who found that group-based interventions enhance peer support and in turn have a positive impact on users’ wellbeing and self-management [33].

Service users expressed increased motivation to attend to the group exercise sessions and felt more motivated to change their behaviour and to initiate exercise. This finding supports the work of other studies which report that peer support and supportive relationships with healthcare professionals enhance motivation to attend intervention sessions and to initiate healthy eating and exercise [27,30,31,34,35].

Service users stressed their need for follow-up with healthcare professionals after the completion of the programme. This request possibly indicates concern about how they would maintain the behaviour change that they achieved over time. Systematic reviews and meta-analyses support the idea that follow-up contact is a critical component of sustainable behaviour change after intervention completion [36,37]. The follow-up process involves evaluation of weight loss along with possible future actions or planning as needed. Given the chronic and complex nature of obesity, extended care may be necessary to maintain a healthy weight in the long term. Consequently, service users’ need for further support to continue to self-monitor their food intake and physical activity and maintain a healthy weight is well-justified. Perri and Ariel-Donges (2018) highlight that successful long-term management of obesity requires long–term maintenance programmes in addition to follow-up appointments [38].

The quality of communication between the user and the care team is a central element of the user-centered care experience in integrated care services. [39]. In this study, the participants stressed the importance of health professionals who are active listeners and able to respond accordingly. There is evidence in the literature which confirms the importance of empathy, cooperation and sufficient time from health care professionals for service users in integrated care settings [40,41,42]. In the literature, supportive healthcare professionals have been described as an essential component of weight management programmes by service users [30,31,34,43]. Friendliness and approachability [27,30,34], being able to communicate verbally [34], listening [27,34], and being encouraging [30] are important aspect of creating trusted relationships with healthcare professionals.

The interviews in this study were conducted during the initial outbreak of the COVID-19 pandemic. Almost overnight, the service was forced to deliver interventions online because of social distancing and national lockdown measures. Service users who had to follow the online mode of delivery of the weight management programme reported that while an online course may provide an information base, it could not answer some of the service users’ questions. A previous qualitative review highlighted how service users preferred the face-to-face support provided by healthcare professionals in weight management programmes and the supportive environment of sharing experiences with peers [44]. In the present study, service users did not comment negatively about the quality of the online mode of delivery; rather, they stressed only a communication issue in that questions they had could not be answered as desired. Therefore, their preference for a face-to-face mode of delivery can be explained by face-to-face sessions allowing social interaction with people with similar experiences and healthcare professionals able to instantly offer an answer to a particular question. Further research is needed to show why service users prefer face-to-face support in a weight management intervention. Findings on this question could inform future policy on weight management delivery specification considering service users’ needs and satisfaction.

In terms of the effectiveness of the programme, service users reflected that they experienced positive health outcomes, improved social lives, and a more optimistic view of life. Specifically, this study found that face-to-face group classes were effective in increasing physical activity. This is in agreement with the previous research in this area, which supports the effectiveness of both face-to-face and remotely delivered interventions in increasing physical activity [45,46,47], although the present study did not find that the online mode of delivery reduced physical inactivity. This is consistent with previous research which supporting the success of online weight management interventions in the stabilization of weight, though not in altering dietary and physical activity behaviour [48].

Overall, the present study indicates that Total Wellbeing Luton embraces user-centredness and supports positive behavior change. In addition, service users commented positively on the flexible referral process for accessing the service and the social support that they received from peers and healthcare professionals. Service users stressed the positive impact that they noticed on their physical and emotional health following their engagement with the service. Areas of improvement in the service have also been identified. The advertisement strategy of the service requires improvement in order to be more ‘visible’ to the public. Furthermore, the lack of allocated time to follow up with service users who have completed a programme has been identified as another area which requires attention.

The COVID-19 pandemic, may have further influenced service users’ experiences. For example, service users highlighted the importance of social support from peers and healthcare professionals and their need to be followed up at the end of the programme. One possible explanation for this could be that the social isolation that service users experienced due to government’s restrictions increased their need to be socially supported. Similarly, it can be suggested that service users’ preference for face-to-face support instead of online support was due to the period of social isolation they had experienced, which may have significantly impacted individual perceptions of loneliness.

The findings of these interviews offer a significant contribution to understanding which aspects service users value the most in an integrated healthcare setting. Service users highlighted the importance of feeling supported, respected, and listened to in a space where their voice is heard and their opinion matters. They value peer support, trust relationships with healthcare professionals, and the self-management skills that they have learned. Furthermore, areas of improvement have been highlighted. Improvements in marketing strategy and advertising to increase awareness about the programme is required. GP surgeries may not be aware of the service, and it was felt that there was inadequate ‘exposure’ of the service to people in Luton for the promotion of self-referral.

These findings should be interpreted within the consideration of certain limitations. Eleven of the fourteen interviewed service users belonged to a “White–British” ethnic background. In the literature, it is documented that ethnicity as a variable can impact resources and policy [49]. It has to be acknowledged that the findings in the present study have been obtained by participants who predominantly belong to a “White–British” ethnic background. The study population is not a representative sample of Luton’s population. According to the last available sociodemographic data provided by the 2011 Census, under half of Luton’s residents (44.6%) belong to a “White–English” ethnicity background [50]. In addition, the small sample size is a limitation of this study; however, data saturation was met. Therefore, the generalisability of the study’s findings is limited due to the small sample size and the lack of diversity in the sample. In addition, only a single setting in a single location in the UK was examined. Future studies with a more diverse population and in multiple settings may provide more representative insights.

Another important consideration in terms of the interpretation of the findings is the wide age range of the participants. Nine of the fourteen participants were over 50 years of age, and it is reasonable to conclude that the findings represent the experiences of an older population rather than the total population who used the service; however, there were no differences identified in user experiences on the basis of age. For example, both participants aged under and over 50 years commented on their need for follow-up and on their preference for engaging in face-to-face sessions.

Despite these limitations, this study provides useful insights into the implementation of a weight management programme within an integrated care context. The findings can be used to improve the integration of care in Total Wellbeing Luton and to inform the implementation of similar programmes.

## 5. Conclusions

The present study demonstrates that service users who attend a weight management programme in an integrated service that also covers mental health, value the ongoing support they receive from the service providers, their camaraderie with fellow users, and being listened to by healthcare professionals. Service users reported changes in their health, and stated that they gained deeper knowledge and the motivation and opportunity to support themselves and to change their behaviour after their participation in the programme. Some dissatisfaction and difficulties were reported concerning the marketing strategies of the service and the online mode of delivery, which was adopted for social distancing purposes due to COVID-19. In order to improve integrated obesity care, service providers need to look beyond organisational integration and consider what service users value the most about the care they receive.

This study makes an important contribution to understanding what is happening in Total Wellbeing through the lenses of service users. In addition, the findings may provide valuable indications to service providers on how to improve existing services, and may be of international interest as countries aim to evaluate the effectiveness of integrated care initiatives and their impact.

The present study was conducted during the COVID-19 pandemic, and the majority of the participants were aged over 50 years. Recognizing that a ‘one-size-fits-all’ approach is unlikely to be appropriate when it comes to integrated care delivery, future studies could consider for which groups of patients and under what circumstances such programmes are implemented and successful. Further research is needed to investigate why service users may prefer face-to-face support in a weight management intervention, as many services have now adapted to continue providing remote delivery.

## Figures and Tables

**Figure 1 ijerph-19-00817-f001:**
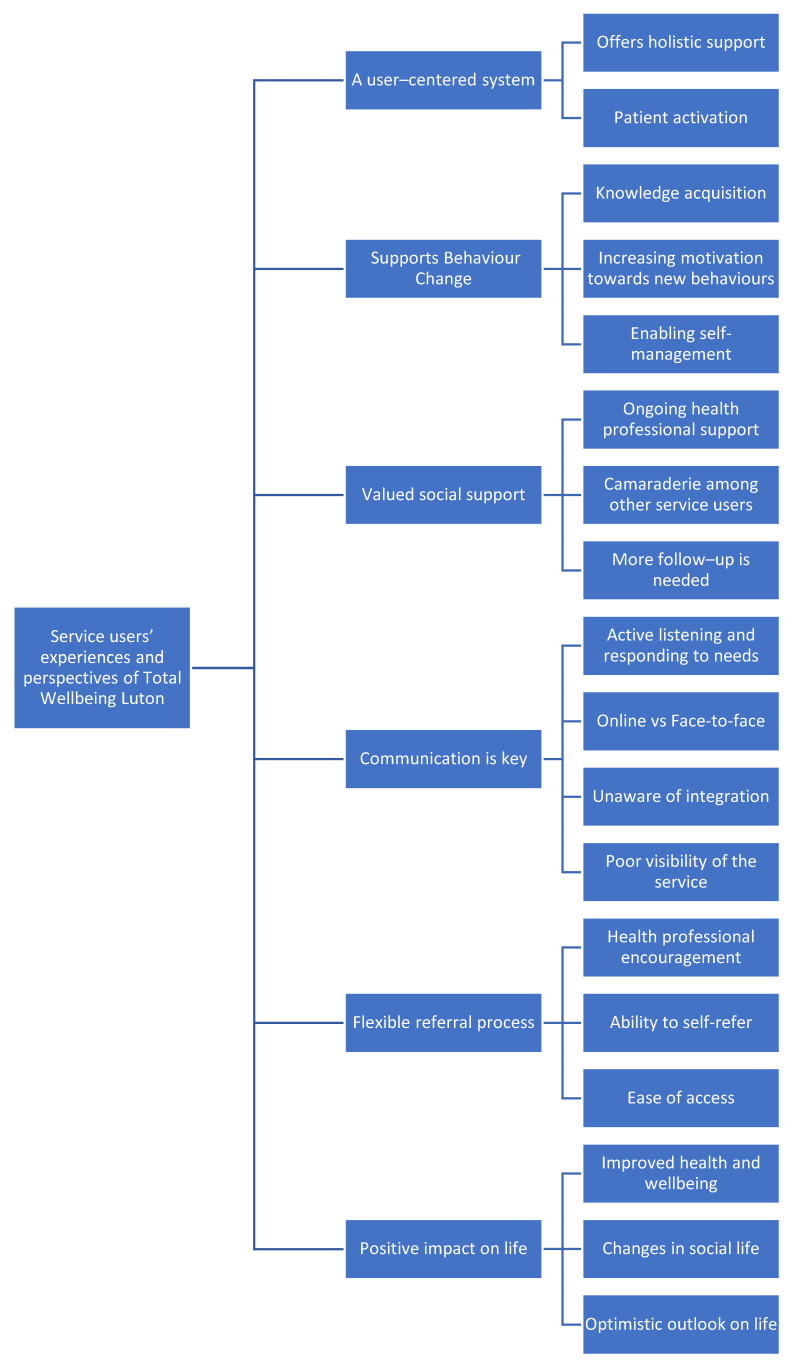
Coding tree of the key themes identified from the data to evaluate service users’ experiences and perspectives of Total Wellbeing Luton.

**Table 1 ijerph-19-00817-t001:** Service users’ characteristics.

Service User ID	Gender	Age	Ethnicity	Type of Referred Services	Status in the Programme (at the Stage of the Interview)
Service user 1	Female	59	White—British	IAPT/smoking cessation/WM	Finished
Service user 2	Female	24	White—British	IAPT	Finished
Service user 3	Female	33	White—British	IAPT	Finished
Service user 4	Male	43	White—Irish	IAPT	In process
Service user 5	Female	59	N/A	WM	Finished
Service user 6	Female	35	Asian/Asian British—Bangladeshi	WM/IAPT/SP	In process
Service user 7	Female	59	White—British	WM/IAPT	Finished
Service user 8	Male	56	White—British	WM	Finished
Service user 9	Male	66	White—British	WM	Finished
Service user 10	Female	41	N/A	WM	Finished
Service user 11	Female	66	White—British	WM	Finished
Service user 12	Female	72	White—British	WM	In process
Service user 13	Male	65	White—British	WM	In process
Service user 14	Male	65	White—British	WM	In process

## Data Availability

No new data were created or analysed in this study. Data sharing is not applicable to this article.

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
