# Peer review of "Understanding the Experience of Service Users in an Integrated Care Programme for Obesity and Mental Health: A Qualitative Investigation of Total Wellbeing Luton"

_ijerph, 2022, doi:10.3390/ijerph19020817_

Round 1

Reviewer 1 Report

General comment

This is a well-written paper that is based on the Total Wellbeing Luton. My main concern is the un-fulfilment of the stated aims. The stated intentions of the proposed study are to first, evaluate the implementation of Total Wellbeing Luton; second, evaluate the experiences of services users on the weight management programme. However, there is no clear description of the program, therefore the reader does not know what type of intervention the participants were presented with. As such, it is unclear how or what are they going to evaluate. Expansion of the introduction with a description of the program with the benefits will allow for a better understanding. Furthermore, the methodology described is aimed to accomplish the second aim, “evaluate the experiences of services users”, however in regards to the first part of the aim, “evaluate the implementation of the program”, the methodology described is not adequate and does not fulfil this part of the aim of evaluating the implementation of the program. For this part to be complete, the use of implementation science including different theories, models and frameworks should be included. As such, reconsideration of the aims of the study should be addressed to make sure they align with the presented methods and results.

Specific comments

In general, the introduction is informative but it does contain repetitive information. Consider shortening, being more specific and including a description of the program and its benefits.

There is no reflexivity of the authors and this should be included in the methods. Include brief reflexivity of the main interviewer author and of the other independent authors who were involved in data analysis.

The age range of the participants is quite wide. Since the sample is small, it is important to understand if the perceptions presented represent an older population or vice versa. For example, 9 out of the 14 participants were above 50 years. Was there a difference in their perceptions based on their age range?

The quality of the qualitative data processing is relatively good; however, the organization of the results in the text is too long. It would benefit to present a brief description of the core six themes in the description and add the sub-themes description and example quotes into tables for the results to flow better.

Being more critical in the discussion, highlighting more in detail the benefits and limits shown by the program itself and how they reflect participants’ experiences including how the COVID-19 pandemic might have influenced the response of participants.

Author Response

Dear reviewer

We are grateful for your time and constructive comments on our manuscript. We have implemented their comments and suggestions and wish to submit a revised version of the manuscript for further consideration in the journal. Below, we provide a point-by-point response explaining how we have addressed each of your comments. We look forward to the outcome of your assessment.

Yours sincerely,

Fani Liapi 

Reviewer 2 Report

The authors have written a manuscript entitled “Understanding the experience of service users in an integrated care programme for obesity and mental health: A qualitative investigation of Total Wellbeing Luton”. Overall a very interesting study and read. It was well written. Please see my suggestions below for improvement.

Abstract

Remove the words background, method, results and conclusion. You will write the content without the headings.

Please consider rewording this sentence: “ These themes evidence how the service operated, what works well, what areas need improvement and what the service users valued the most about the received care.”

Keywords

Please rearrange in alphabetical order.

Introduction

Please add a reference to the first sentence.

WHO is World Health Organization (organization in this instant should have a Z).

Please can you use a more latest reference for this sentence “The World Health Organization (WHO) states that worldwide obesity has tripled since 1975 (1).”

When you use the term healthy weight what are you referring to? Is it normal weight?

Instead of saying living with excess weight you should rather only use the terms overweight and obesity.

Need to refine this section a little more to emphasis the importance of the study.

Methods

Study sample: how was the n calculated. Please give more details on your sample selection methodology. Did you pilot your questions?

Data collection: add supplementary information (S1: Interview schedule for service users)

e.g. Individual semi-structured interviews (Supplementary information S1: Interview schedule for service users)  were conducted. [Make sure you label your supplementary information so the reader will know when to refer to it or the fact that it is available]

Did the participants email their signed consent? Was consent confirmed during the interview? How was the phone interview recorded?

Data analysis:  you do not need to name the researcher in your manuscript. You rather should use the term principle investigator throughout where applicable. This comment applies to other initials in the manuscript. Rather use terms like another researcher etc.

Figure 1 is a really good illustration.

In your method explain the COVID-19 situation and what was allowed at the time.

Need to give a little more detail under the methods section.

Results

Heading 3 should be called results instead of findings.

It would be useful to include n values in your results sections. Suggestions: sometimes it is useful to present thematic analysis in a table.

Be consistent with how you write COVID-19. Please check throughout.

What is SU1? SU6?

When you talk about majority, how many out of the 14 had that perspective?

Discussion

You mention “Further research on this aspect is warranted” elaborate a little further. Why is it warranted?

Conclusion

The way forward and implications should be presented in the conclusion.

General

Try and keep the way you write face-to-face consistent throughout.

Ensure you follow the format of the journal consistently. 

Author Response

(The authors gave the same response as above.)
